# Gut–Brain Axis in Mood Disorders: A Narrative Review of Neurobiological Insights and Probiotic Interventions

**DOI:** 10.3390/biomedicines13081831

**Published:** 2025-07-26

**Authors:** Gilberto Uriel Rosas-Sánchez, León Jesús Germán-Ponciano, Abraham Puga-Olguín, Mario Eduardo Flores Soto, Angélica Yanet Nápoles Medina, José Luis Muñoz-Carillo, Juan Francisco Rodríguez-Landa, César Soria-Fregozo

**Affiliations:** 1Programa de Estancias Posdoctorales por México, Secretaría de Ciencia, Humanidades, Tecnología e Innovación SECIHTI, Centro Universitario de Los Lagos, Universidad de Guadalajara, Lagos de Moreno 47460, Jalisco, Mexico; 2Centro Universitario de Los Lagos, Universidad de Guadalajara, Lagos de Moreno 47460, Jalisco, Mexico; 3Laboratorio de Neurofarmacología, Instituto de Neuroetología, Universidad Veracruzana, Xalapa 91190, Veracruz, Mexico; lgerman@uv.mx (L.J.G.-P.); juarodriguez@uv.mx (J.F.R.-L.); 4Programa Investigadoras e Investigadores por México, SECIHTI-Centro de EcoAlfabetización y Diálogo de Saberes, Universidad Veracruzana, Xalapa 91060, Veracruz, Mexico; abpuga@uv.mx; 5Laboratorio de Neurobiología Celular y Molecular, División de Neurociencias, Centro de Investigación Biomédica de Occidente (CIBO), Instituto Mexicano del Seguro Social, Sierra Mojada 800, Independencia Oriente, Guadalajara 44340, Jalisco, Mexico; mariosoto924@yahoo.com.mx (M.E.F.S.); angelica.napoles@alumnos.udg.mx (A.Y.N.M.); 6Laboratorio de Inmunología, Centro Universitario de los Lagos, Universidad de Guadalajara, Lagos de Moreno 47460, Jalisco, Mexico; mcbjlmc@gmail.com

**Keywords:** anxiety, depression, *Lactobacillus*, *Bifidobacterium*, anxiolytic, antidepressant, probiotics, MGB Axis

## Abstract

The gut microbiota and its interaction with the nervous system through the gut–brain axis (MGB) have been the subject of growing interest in biomedical research. It has been proposed that modulation of microbiota using probiotics could offer a promising therapeutic alternative for mood regulation and the treatment of anxiety and depression disorders. The findings indicate that several probiotic strains, such as *Lactobacillus* and *Bifidobacterium*, have demonstrated anxiolytic and antidepressant effects in pre and clinical studies. These effects seem to be mediated by the regulation of the hypothalamic–pituitary–adrenal axis (HPA), the synthesis of neurotransmitters such as serotonin (5-HT) and Gamma-amino-butyric acid (GABA), as well as the modulation of systemic inflammation. However, the lack of standardization in dosing and strain selection, in addition to the scarcity of large-scale clinical studies, limit the applicability of these findings in clinical therapy. Additional research is required to establish standardized therapeutic protocols and better understand the role of probiotics in mental health. The aim of this narrative review is to discuss the relationship between the gut microbiota and the MGB axis in the context of anxiety and depression disorders, the underlying neurobiological mechanisms, as well as the preclinical evidence for the effect of probiotics in modulating these disorders. In this way, an exhaustive search was carried out in scientific databases including PubMed, ScienceDirect, Scopus, and Web of Science. Preclinical research evaluating the effects of different probiotic strains in animal models during chronic treatment was selected, excluding those studies that did not provide access to the full text.

## 1. Introduction

Alterations in the emotional and affective state such as anxiety and depression disorders have become the leading cause of disability worldwide and represent a public health problem with a growing prevalence in the world’s population. According to the World Health Organization (WHO), depressive disorders affect more than 280 million people in the world, while anxiety disorders have a prevalence of 4% globally [1]. These conditions are associated with a significant burden of disability and a decrease in patients’ quality of life. Despite the availability of pharmacological and psychological therapies, a considerable percentage of patients do not respond adequately to conventional interventions or experience undesirable side effects [2,3].

These disorders are characterized by a considerable burden of disability, significant functional impairment, and a marked reduction in the patient’s quality of life, affecting not only the individual but also the family and social environment [4]. The etiopathogenic complexity of these disorders involves multiple neurobiological factors, including alterations in neurotransmitter systems, dysregulation of the HPA axis, neuroinflammatory processes, and genetic and epigenetic factors [5,6]. Despite advances in the development of pharmacological and psychological therapies, a significant percentage of patients, estimated at 30–40%, do not respond adequately to conventional interventions or experience adverse side effects that limit adherence to treatment [7,8].

In this context, there has been a growing interest in the MGB axis as one of the key pathways for mental health modulation. This axis describes the two-way communication between the central nervous system (CNS) and the gastrointestinal system, where the gut microbiota plays a significant role in the regulation of neuroinflammatory processes, neurotransmitter production, and stress response by modulation of the HPA axis [4,5]. Recent scientific findings have shown that the gut microbiota can directly influence brain function through several pathways: the production of bioactive metabolites; the modulation of intestinal permeability; the synthesis of neurotransmitters such as 5-HT, GABA, and dopamine (DA); and the regulation of the immune and inflammatory response [6,7].

Recent evidence suggests that gut dysbiosis, or the imbalance in the composition of the microbiota, may contribute to the development of psychiatric disorders such as depression and anxiety. Experimental studies have shown that gut dysbiosis can induce depression-like behaviors through abnormal synaptic pruning mechanisms mediated by microglia and regulated by complement C3 [9]. Individuals with depressive and anxiety disorders often exhibit changes in the alpha diversity of the gut microbiota, with a marked reduction in beneficial bacterial genera such as *Lactobacillus* and *Bifidobacterium* and an increase in pro-inflammatory species [8,10]. In this framework, probiotics have gained attention as an emerging therapeutic strategy to improve mental health. It has been proposed that these beneficial bacteria specifically from the genera *Lactobacillus* and *Bifidobacterium* can modulate gut microbiota; reduce inflammation; and alter the production of neurotransmitters involved in mood regulation, such as 5-HT and GABA [11,12].

Preclinical and clinical studies have documented the anxiolytic and antidepressant effects of some probiotic strains administered individually or combined, although the precise mechanisms of action are not yet fully understood [13]. Recent randomized controlled trials have shown that short-term, high-dose probiotic supplementation in depressed patients can reduce depressive symptoms and microbiological changes in the gut and nervous system [14]. However, the exact mechanisms of action, optimal dosage, duration of treatment, and strain-specific effects are not yet fully understood, so further research is needed to establish standardized treatment protocols. The heterogeneity of study designs, populations studied, probiotic strains used, doses administered, and clinical assessment tools has led to some variability in results, although the general trend suggests positive effects [9,15].

The present narrative review aims to discuss the relationship between the gut microbiota and the microbiota–gut–brain axis in the context of anxiety and depression disorders. The underlying neurobiological mechanisms linking gut dysbiosis to CNS dysfunction will be investigated, as well as the preclinical evidence for the effect of probiotics in modulating these disorders. Their mechanisms of action and potential clinical applications will be highlighted.

## 2. Method

### 2.1. Design

This study was designed as a narrative literature review on the effect of probiotics on anxiety and depression, analyzing information from scientific papers, book chapters, books, and official websites to create an informative, critical, and useful synthesis on the topic. In the narrative review, there are no predetermined research questions or a specific search strategy, only a topic of interest. Recognizing that narrative reviews lack systematic methods for identifying, appraising, and synthesizing information, which could lead to authors including or excluding information to support a particular position [16,17], we describe the parameters that were used to include or exclude studies to ensure objective inclusion of information. These included conducting a search and identifying keywords, reviewing abstracts and articles, and documenting results [18], as described below.

### 2.2. Article Criteria

The inclusion criteria focused on research articles, reviews, and position statements examining the effects of different probiotic strains in preclinical and clinical studies during chronic treatment and their mechanism of action in English-language articles. The decision to include only studies with chronic treatment is because the positive effects of probiotics need time to become established and clinically manifested. Probiotics need to colonize the gut, modulate the existing microbiota, and exert their immunomodulatory and metabolic effects, processes that do not occur with acute or short-term administration. In addition, most conditions for which probiotics are studied (e.g., irritable bowel syndrome, inflammatory bowel disease, or improvement of immune function) are chronic and require long-term interventions. Exclusion criteria were studies that did not provide access to full text, unofficial websites, duplicate publications, and dissertations.

### 2.3. Article Research

Data on the topic described in the inclusion criteria were searched in specialized databases such as PubMed, Science Direct, Web of Science, and Scopus using a combination of specific words such as “*anxiety*”, “*depression*”, “*neurobiology*”, “*MGB axis*”, “*anxiolytic*”, “*antidepressant*”, “*probiotics*”, “*treatment*”, “*animal*”, “*models*”, “*clinical studies*”, and “*probiotic mechanism of action*”.

## 3. General Aspects of the Microbiota and Its Relationship with the Microbiota–Gut–Brain Axis

The human microbiota, a complex microbial ecosystem residing primarily in the gastrointestinal tract, comprises approximately 38 trillion (1012) microorganisms, a figure comparable to the number of human cells (ratio ~1.3:1) [19]. The human gut, in particular, harbors around 100 trillion microorganisms, including bacteria, archaea, fungi, and viruses, constituting the intestinal microbiota [20]. This dense and diverse microbial population plays a crucial role in host homeostasis, contributing to essential physiological processes. The intestinal microbiota actively participates in digestion, facilitating the breakdown of complex macromolecules and nutrient absorption; the biosynthesis of vitamins, such as vitamin K and some B vitamins; and maintaining the integrity of the intestinal barrier mucosa, protecting the organism against pathogen invasion and toxin translocation. Furthermore, it bidirectionally interacts with the immune system, modulating its responses and contributing to organismal homeostasis [21,22].

The intestinal microbiome is characterized by remarkable interindividual diversity, harboring a wide range of bacterial species. The phyla *Bacteroidetes* and *Firmicutes* predominate in the composition of the intestinal microbiota, followed by *Actinobacteria*, *Proteobacteria*, and *Verrucomicrobia* in smaller proportions [14]. Recent research has revealed a correlation between nationality and composition, as well as the functionality, of the intestinal microbiome, suggesting the influence of geographical, dietary, and cultural factors on the structure and function of this microbial community. These differences can influence the diversity and abundance of microbial species in the gut as well as their metabolic activities [16,23,24].

### 3.1. The Microbiota–Gut–Brain Axis

The MGB axis describes the complex bidirectional communication between the brain and the gut microbiota. This interaction is established through multiple pathways, including the endocrine HPA axis and immunological, neural (vagal nerve afferent and efferent pathways, sympathetic nervous system, and CNS), and metabolic pathways. The integrity of the intestinal and blood–brain barriers (BBB), as well as bacterial production of neurotransmitters and metabolites (tryptophan, 5-HT, short-chain fatty acids [SCFAs]), are crucial for the correct function of the MGB [17,18,24]. The MGB integrates brain signals that modulate the sensory, secretory, and motor functioning of the gut. Reciprocally, signals and metabolites derived from the gut microbiota influence brain biochemistry, behavior, development, and function. It is important to highlight that these communication mechanisms are not mutually exclusive and can interact with each other, generating a complex network of interactions that influence brain function and behavior (Figure 1).

### 3.2. Hypothalamic–Pituitary–Adrenal Axis

The HPA axis plays a crucial role in the bidirectional communication of the MGB axis. In response to stressors, the brain activates the HPA axis. The hypothalamus releases corticotropin-releasing hormone, which stimulates the release of adrenocorticotropic hormone by the pituitary gland. Adrenocorticotropic hormone (ACTH), in turn, induces the release of cortisol by the adrenal glands. Cortisol, a stress hormone, can affect gut barrier function, intestinal motility, and microbiota composition [25]. The gut microbiota interacts with the HPA axis through various mechanisms. Dysbiosis increases the expression levels of pro-inflammatory cytokines (IL-1β, IL-6, TNF-α) that activate the HPA axis, and bacterial lipopolysaccharides and peptidoglycans also contribute to this activation. SCFAs, upon crossing the BBB, reduce microglial activity and HPA axis activity. Bidirectional communication between the microbiota, the CNS, and the HPA axis occurs through the enteric nervous system and the vagus nerve. The microbiota produces neurotransmitters such as acetylcholine, DA, GABA, histamine, norepinephrine (NE), and 5-HT and influences catecholamine transport and tryptophan availability [26,27].

### 3.3. Vagus Nerve

The vagus nerve plays a fundamental role in the complex bidirectional communication that characterizes the MGB axis. This communication is established by transmitting sensory information from the gut to the brain through afferent fibers (80%) and modulating intestinal functions from the brain via efferent fibers (20%) [28]. Anatomically, the vagus nerve extends from the brainstem to the abdomen, innervating the gastrointestinal tract, including the enteric nervous system, often called the “second brain” due to its intricate neuronal network. The afferent fibers of the vagus nerve, with endings in the muscle and mucosal layers of the intestine, act as sensors, detecting both mechanical stimuli, such as changes in intestinal volume, and chemical stimuli. The latter includes a variety of signaling molecules, including neurotransmitters, hormones, and cytokines, whose concentration can be modulated by the gut microbiota [29,30].

The gut microbiota’s influence on vagal signaling occurs through various mechanisms. In addition to directly modulating the concentration of neurotransmitters, hormones, and cytokines in the gut, the gut microbiota can induce the release of these molecules from immune and enteroendocrine cells [31,32]. Likewise, certain intestinal bacteria have the ability to produce neurotransmitters, such as 5-HT, which, after absorption, can exert a direct influence on vagal afferent nerve endings or indirectly modulate the activity of enteric neurons, thus contributing to gut–brain communication [33,34]. For example, the bacterium *Lactobacillus gasseri* can modulate the inflammatory response in the gut and brain, reducing IL-1β levels and improving cognitive function and behavior in animal models through the vagus nerve. This complex interaction highlights the importance of the vagus nerve as a key component in regulating physiological and behavioral functions [35].

### 3.4. Intestinal Permeability and Blood–Brain–Barrier Permeability

The intestinal epithelial barrier, composed of microbial biofilms, the mucus layer, the gut-associated immune system, and the epithelial monolayer, regulates the entry of nutrients and restricts the passage of harmful substances, such as bacterial lipopolysaccharides, into the bloodstream [36]. The disruption of this barrier, a phenomenon known as “leaky gut” or “intestinal hyperpermeability,” has been associated with some pathological conditions, including neurological and psychiatric disorders [37,38]. Dysbiosis compromises this barrier through several mechanisms: the reduction in the production of SCFAs, which are crucial for the integrity of intercellular tight junctions; the increase in pro-inflammatory metabolites that damage epithelial cells and tight junctions; and bacterial translocation, including Gram-negative bacteria and lipopolysaccharides (LPS), which induce systemic inflammation and intestinal barrier dysfunction [20,39,40]. A study by Wu et al. [41] provides evidence for the involvement of intestinal hyperpermeability in major depressive disorders in adolescents. In this study, biomarkers such as zonulin, intestinal fatty acid-binding protein (I-FABP), LPS, and claudin-5 were identified, which could be useful for the diagnosis and monitoring of this condition.

The BBB and the intestinal barrier share molecular and cellular similarities; furthermore, neuroglial cells play a key immunological role in the integrity of the blood–brain barrier. Studies in mice show that antibiotic-induced microbiota depletion decreases the expression of tight junction proteins in the BBB, increasing its permeability, and fecal microbiota transplantation restores the integrity of the BBB and the expression of these proteins, confirming the relationship between intestinal dysbiosis and BBB dysfunction [42].

### 3.5. Immunological Pathway

The intestinal microbiota plays a crucial role in the development and maturation of gut-associated lymphoid tissue and in maintaining intestinal homeostasis [31,32,43]. Under eubiotic conditions, the microbiota contributes to host defense by inhibiting the colonization of pathogens through the production of IgA and antimicrobial peptides. CD103+ dendritic cells play a crucial role in intestinal immunity. These cells release IL-10, an immunomodulatory cytokine that influences the differentiation of T and B cells. In addition, CD103+ dendritic cells participate in the conversion of dietary vitamin A to retinoic acid, a process enhanced by the presence of SCFAs, a product of bacterial fermentation of dietary fiber. This local production of retinoic acid is essential for the induction of immunological tolerance and the regulation of the immune response in the gut [18,44,45].

Conversely, dysbiosis facilitates the translocation of harmful metabolites and pathogens, recognized by pattern recognition receptors, into the systemic circulation, contributing to intestinal inflammation. The activation of pattern recognition receptors triggers the activation of intracellular signaling pathways, particularly NF-κB, leading to the release of pro-inflammatory cytokines such as IL-1, IFNγ, and TNF-α, further exacerbating intestinal inflammation [46,47].

### 3.6. Microbial Metabolites

Microbial metabolites, products of the metabolism of the gut microbiota, play a fundamental role in the bidirectional communication between the gut and the brain, modulating brain function and behavior. This complex communication network is influenced by a variety of metabolites, including neurotransmitters, secondary bile acids, and SCFAs. Alterations in the composition and function of the gut microbiota, known as dysbiosis, can affect the production and profile of these metabolites, contributing to the development of neuropsychiatric disorders such as depression and anxiety [48].

### 3.7. Short-Chain Fatty Acids

Short-chain fatty acids are crucial metabolites derived from the fermentation of undigested dietary fiber and non-digestible carbohydrates, playing a fundamental role in modulating the MGB axis. SCFAs are saturated fatty acids containing one to six carbon atoms in their aliphatic chain with a carboxylic acid group, occurring in both linear and branched conformations [49]. The bacterial genera capable of producing SCFAs as a byproduct in the gut are *Lactobacillus*, *Akkermansia*, *Ruminococcus*, *Ruminococcus*, *Blautia*, *Faecalibacterium*, *Bacteroides*, *Lactobacillus*, *Bifidobacterium*, *Lacticaseibacillus*, *Ligilactobacillus*, *Prevotella*, *Enterococcus*, *Eubacterium*, *Roseburia*, *Fusicatenibacter*, *Clostridium*, and *Coprococcus*. It is important to mention that the amount and proportion of SCFAs generated depend not only on the composition of the microbiome and the quantity of microorganisms present in the gut but also on the dietary fiber that serves as a substrate for fermentation. The most common SCFAs are acetic acid, propionic acid, and butyric acid (in a molar ratio of approximately 3:1:1) [50].

These SCFAs exert pleiotropic effects both locally and systemically, impacting intestinal health and brain function. At the intestinal level, they regulate the function and integrity of the intestinal barrier through mucin synthesis. They also regulate the activity of the histone deacetylase complex, which affects the transcription of transcription factors that promote inflammatory processes such as nuclear factor-κB. SCFAs promote the differentiation of regulatory T cells and the production of anti-inflammatory cytokines such as interleukin-10 in addition to regulating the activity and activation of the NLRP3 inflammasome [51]. At the endocrine level, SCFAs promote the secretion of intestinal hormones such as glucagon-like peptide-1 (GLP-1) and peptide YY (PYY). GLP-1 plays a crucial role in regulating appetite and glucose metabolism. Secreted by L cells in the small intestine, this hormone increases in response to food intake. This peptide is also produced in the brain and has been shown to play a critical role in regulating neurogenic processes in addition to decreasing inflammation and apoptosis [52].

Moreover, PYY has been shown to exert neuroprotective effects, decreasing oxidative stress by inhibiting amyloid-β peroxidation-activated lipids and modulating brain-derived neurotrophic factor, a protein with a role in neuronal proliferation, differentiation, and survival [53]. Butyric acid constitutes the main energy source for colonocytes and contributes significantly to maintaining the integrity of the intestinal epithelial barrier. This function is essential for preventing bacterial translocation and systemic inflammation, factors implicated in the pathogenesis of neuropsychiatric disorders, including depression and anxiety [54,55].

Likewise, propionic acid and acetic acid, two others relevant SCFAs, demonstrate neuromodulatory activity, affecting the morphology and activation of microglial cells, thus preventing the death of neuronal cells. Furthermore, it has been shown that propionic acid and acetic acid can cross the BBB via monocarboxylate transporters and influence the integrity of the barrier by promoting the expression of tight junction proteins [53]. After crossing the BBB, these metabolites influence the function of glial and neuronal cells. Acetic acid, for example, can stimulate the production of neurotransmitters such as GABA and 5-HT, contributing to neurochemical homeostasis and potentially mitigating depressive states [56]. Propionic acid exerts anxiolytic and anti-inflammatory effects in the brain by modulating specific receptors, such as free fatty acid receptor 3 [57]. Preclinical evidence supports the impact of diet on modulating SCFA levels and their influence on behavior. Diets rich in fiber, by increasing SCFA production, attenuate depression and anxiety-like behaviors in animal models. These beneficial effects are attributed to both peripheral mechanisms, such as the decrease in pro-inflammatory cytokines, as well as central mechanisms, including the activation of vagal signaling pathways and the epigenetic regulation of genes related to the stress response [58,59].

## 4. Neurobiology of Anxiety and Depression

Currently, anxiety and depression have become neuropsychiatric disorders with the highest incidence and prevalence worldwide and represent a major public health problem. According to the WHO, depressive disorder affects approximately 380 million people worldwide, making it one of the leading causes of disability worldwide [60]. On the other hand, pathological anxiety affects more than 300 million people. Both disorders exhibit marked sexual dimorphism, meaning they are more prevalent in women than in men. After the COVID-19 pandemic, anxiety and depression increased by up to 25% during the first year of the pandemic [61].

Both disorders have multifactorial etiology, caused by genetic, environmental, psychological, and social factors. Genomic studies have reported genetic loci and gene variations associated with alterations in dopaminergic, noradrenergic, serotonergic, and GABAergic neurotransmission; i.e., low levels of these neurotransmitters are observed [62]. On the other hand, chronic stress is considered a crucial precipitating factor in the development of emotional disorders because the activation of the HPA axis can promote changes in the function and structure of brain regions such as the hippocampus, septum, amygdala, raphe, and cerebral cortex, structures of the limbic system responsible for regulating emotional and affective states [63]. Therefore, exposure to stress at an early age increases the risk of developing signs and symptoms of anxiety and depression in adulthood [64].

Neurobiological etiology of anxiety and depression disorders include alterations in neurotransmission systems and neuronal circuits in brain structures responsible for modulating emotional state. Neurotransmission systems implicated in both disorders include 5-HT, NE, DA, and GABA [65], among others. The monoaminergic hypothesis of depression provided the basis for the development of antidepressants such as tricyclic antidepressants (TCAs) and monoamine oxidase inhibitors (MAOIs), in addition to current antidepressants such as selective 5-HT reuptake inhibitors (SSRIs) and 5-HT and NE reuptake inhibitors (SNRIs), which increase synaptic monoamine concentrations in the short term, thus reversing the signs and symptoms of anxiety and depression [66]. The monoaminergic hypothesis of depression is based on the molecular mechanisms of action induced by currently available antidepressants. However, approximately 33% of patients responded clinically to these agents, while 33% showed placebo-like responses [67]. Although researchers strongly suggest that the mechanisms involved in depression are not solely related to monoamines, this theory remains relevant to explaining certain aspects of the disease; however, further study of other pathogenetic factors is needed

Neuroimaging studies have identified that in anxiety and depression, consistent alterations are observed in key brain regions. These involve a complex network of brain structures, including the amygdala, hippocampus, medial prefrontal cortex (mPFC), hypothalamus, midbrain (e.g., raphe nuclei), and brainstem (e.g., periaqueductal gray matter, SGPA), among others, that modulate responses to anxiety and depression [68]. In major depressive disorder (MDD), hypoactivity in the dorsolateral and anterior prefrontal cortex and hyperactivity in the amygdala and subgenual cingulate gyrus have been reported [69]. In anxiety, hyperactivity is reported in the amygdala, insula, and anterior cingulate cortex, added to a malfunction in the fronto-limbic circuits [70].

Currently, stress is considered an event or experience that threatens an individual’s ability to cope and adapt [71]. In response to stress, the body modifies the concentrations of glucocorticoids, catecholamines, growth hormone, and prolactin, whose effect is to adapt the individual to the new condition. These hormones and cellular mediators promote pathophysiological processes when the response is excessive or dysregulated [72]. The HPA axis plays a fundamental role in the pathophysiology of anxiety and depression. Its dysregulation is characterized by high and persistent levels of cortisol in patients with anxiety and major depression [73]. This dysregulation contributes to neuroplastic alterations, including hippocampal atrophy and reduced adult neurogenesis [74]. Glucocorticoids as a cortisol in humans and corticosterone in rodents play a crucial role in the regulation of energy metabolism; stress-related homeostasis; cardiovascular function; and neuroendocrine, immune, and inflammatory responses [75].

In some patients with depression associated with exacerbated activity of the HPA axis, enlargement of the pituitary and adrenal glands, in addition to hypercortisolism, were observed [76]. Currently, chronic stress is known as “the glucocorticoid theory of depression”; the central hypothesis of this theory posits that abnormal activation of the HPA axis and excess cortisol are crucial to the neurobiology of depression. Factors such as monoamine dysfunction, decreased neurogenesis, synaptic neuroplasticity, increased neurodegeneration, and regional brain changes associated with depression have been associated with the glucocorticoid theory of depression [77].

Several studies have included neuroinflammation as a crucial pathophysiological mechanism in disorders of the emotional and affective state. Reports indicate that patients with MDD and anxiety have elevated levels of proinflammatory cytokines such as IL-1β, TNF-α, IL-6, and IFN-γ [78]. The prolonged activation of the immune system influences serotonergic neurotransmission through the activation of the enzyme indoleamine 2,3-dioxygenase (IDO), which diverts tryptophan towards the kynurenine pathway instead of serotonin synthesis, a situation that causes the depletion of 5-HT, a neurotransmitter crucial in the regulation of emotions [79]. Several studies have shown that acute phase proteins, chemokines, and adhesion molecules increased in the peripheral blood of depressed patients [80,81], and it has been observed that the administration of the cytokine interferon alfa (IFN-α) and inflammation inducers such as LPS and typhoid vaccine contributed to behavioral changes like those observed in depressed patients.

Cytokines can affect behavior through alteration of metabolism of NE, 5-HT, and DA as well as neuroendocrine functions, leading to a change in the cortisol curve and elevation of its nocturnal concentrations [81,82]. Proinflammatory cytokines act by activating the enzyme cyclooxygenase-2 (COX-2), which enhances the synthesis of prostaglandins E2 (PGE2), which in turn activates inflammatory cells and induces inflammatory reactions. During the chronic inflammatory process, proinflammatory cytokines increase and anti-inflammatory cytokines decrease, leading to the development of comorbidities [83]. The results of a study involving differences in the levels of proinflammatory cytokines produced by monocytes between patients with depression and healthy controls demonstrated that basal concentrations of IL-6 and IL-1β are significantly increased in depressed subjects compared to the control group [84], suggesting that in depression, immune activity is significantly increased. Antidepressant pharmacotherapies are known to reduce the synthesis of peripheral pro-inflammatory cytokines and induce the production of anti-inflammatory cytokines such as IL-10, Transforming Growth Factor beta-1 (TGF-β1), and brain-derived neurotrophic factor (BDNF), exerting a significant immunoregulatory effect in patients with depression [85,86]. TGF-β1 levels have a negative correlation with the score obtained on the Hamilton Depression Rating Scale [87].

Neuronal plasticity has been demonstrated in several experimental studies, the results revealing that exposure to stress can cause changes in neuronal processes or in the number of neurons as well as atrophy of hippocampal CA3 pyramidal neurons and reduced proliferation in the dentate gyrus [72,88]. In addition, neuroimaging studies in depressed patients revealed selective structural changes in several limbic and non-limbic structures, such as the prefrontal and cingulate cortexes. In these regions, both metabolism and volume are reduced, while hippocampal atrophy occurs with the progression of depressive syndrome [89,90]. Post-mortem morphometric studies have shown a significant reduction of glial cell densities in some cortical and limbic areas of the brain [91]. Therefore, depression may be associated with an impairment of structural plasticity and cellular resilience, and antidepressant pharmacological therapies may act to normalize this impairment [90]. In addition, the adult brain and the developing and maturing brain are influenced by factors such as BDNF, nerve growth factor (NGF), and neurotrophin-3. During episodes of intense stress, BDNF expression is downregulated in neurons of the dentate gyrus, CA3, CA1, and pyramidal cell layers [71], and this downregulation process leads to atrophy of the CA3 neurons and consequently to reduced neurogenesis in hippocampal granule cells. Unlike stress, subchronic treatment with clinically effective antidepressant drugs increases BDNF expression in the hippocampus and frontal cortex [92], and behavioral studies have shown that upregulation of BDNF is an essential factor for the therapeutic action of antidepressants and therefore for the success of therapy [93].

Based on these studies, it is possible to suggest that BDNF is directly correlated with stress, neurogenesis, and hippocampal atrophy during depressive episodes. However, it is important to highlight that cAMP Response Element Binding Protein (CREB) upregulates the BDNF gene and that antidepressant treatments currently used in clinical practice, which increase NA and/or 5-HT concentrations in the synaptic cleft, stimulate CREB expression, and BDNF is therefore closely related to the monoaminergic hypothesis of depression. Importantly, all the cardinal signs of inflammation have been observed in both depression and cardiovascular diseases. For this reason, BDNF is considered a point of convergence between neurotrophins and the “inflammasome” hypothesis in depressive disorders, as this neurotrophin plays a crucial role in brain neurogenesis in affective disorders.

On the other hand, the importance of the MGB axis in the modulation of mood and cognitive processes has recently been evidenced. This axis exerts its brain function through bidirectional communication pathways [6]. These communication mechanisms include the vagus nerve; modulation of the HPA axis; and the metabolic pathway that includes the synthesis of SCFAs and neurotransmitters such as 5-HT, GABA, and DA [33,94]. Clinical studies report alterations in the composition of the intestinal microbiota in patients with MDD and anxiety [95]. On the other hand, it has been reported that administration with probiotics favors the synthesis of butyrate, propionate, and acetate (SCFAs) capable of crossing the BBB and exerting effects on neurogenesis, reducing neuroinflammation, and modulating microglia [96]. Additionally, it has been shown that in patients with MDD, elevated levels of LPS alter intestinal permeability, favoring bacterial translocation and systemic activation of the immune system, contributing to the neuroinflammation present in patients with MDD [97]. This is relevant because 95% of the body’s 5-HT is produced in the intestine, and dysbiosis significantly influences the supply of tryptophan, which is crucial for the synthesis of 5-HT and the regulation of anxiety and depression disorders [98].

## 5. Treatment of Anxiety and Depression in Preclinical Studies

Therapeutic approaches are commonly employed to manage mental health disorders, including anxiety, which, like depression, can significantly impair daily functioning. Anxiety disorders are characterized by excessive worry, fear, and physical symptoms and require a diverse range of treatment options, with traditional antidepressants commonly used in treatment. SSRIs are typically considered the first-line treatment for anxiety, while SNRIs are often used when SSRIs fail or prove ineffective [99]. In contrast, MAOIs and TCAs are less frequently prescribed due to their significant side effects and reduced tolerability [100]. Despite their proven efficacy, antidepressants have a delayed onset of therapeutic effects, making them unsuitable for the acute treatment of anxiety. As a result, benzodiazepines are often employed in these situations, offering rapid anxiolytic effects [101]. The primary mechanism of action of benzodiazepines involves potentiating the activity of GABA on the GABA_A_/benzodiazepine receptor complex. Benzodiazepines facilitate the opening of the chloride ion channels, which increases chloride influx into neurons. This leads to membrane hyperpolarization and a swift reduction in anxiety symptoms. However, the use of benzodiazepines is associated with considerable side effects, including sedation, muscle relaxation, psychomotor impairment, and anterograde amnesia [102], in addition to the development of pharmacological tolerance. Long-term use can also result in dependence, withdrawal symptoms, cognitive impairment, and an increased risk of intoxication, especially when combined with alcohol and other depressors of the CNS. Due to these risks, benzodiazepines are typically prescribed in conjunction with antidepressants, but their use should be carefully managed and gradually tapered to avoid adverse outcomes [103].

Given the short- and long-term side effects of benzodiazepines, antihistamines, which act as histamine-1 (H1) receptor antagonists, have become a common alternative for treating anxiety, panic attacks, and insomnia. These medications are generally well-tolerated, with fewer risks than benzodiazepines, but they may cause dry mouth, constipation, sedation, and impairment in activities such as driving. A major limitation of antihistamines is the development of tolerance over time, which can reduce their long-term effectiveness [104].

Benzodiazepines, while faster acting than antidepressants [101], may not provide sufficient immediacy in certain anxiety disorders, such as social anxiety or panic disorders. In these cases, beta-adrenergic receptor antagonists, or beta-blockers, are employed for their ability to address acute physiological symptoms [105]. These agents act by blocking adrenergic receptors, particularly in the cardiovascular system, thereby inhibiting the effects of catecholamines such as adrenaline. This mechanism reduces tachycardia, lowers blood pressure, and alleviates acute symptoms like palpitations and panic episodes [106]. Beta-blockers are particularly beneficial for managing the somatic manifestations of anxiety in conditions like social anxiety and performance-related phobias. Unlike benzodiazepines, they do not induce sedation or carry a risk of dependence, offering a more favorable safety profile for targeted use. However, beta-blockers are limited in their therapeutic scope as they primarily mitigate physical symptoms and do not address the psychological components of anxiety, such as fear, avoidance behaviors, or pervasive distress. Consequently, their use is often adjunctive, tailored to specific clinical scenarios requiring rapid symptomatic relief [105].

On the other hand, depression is a complex disorder requiring specific therapeutic approaches, and various pharmacological treatments have been developed to address its symptoms. Among the most used are SSRIs, SNRIs, TCAs, and MAOIs, each with distinct mechanisms of action. SSRIs, such as fluoxetine and sertraline, work by blocking serotonin reuptake at neuronal synapses, increasing its availability in the brain, thereby improving the patient’s mood [107]. Meanwhile, SNRIs inhibit the reuptake of both serotonin and norepinephrine, making them effective in a broader range of depressive and anxious symptoms [108]. Although TCAs are older drugs, they remain useful in certain cases as they block the reuptake of serotonin and norepinephrine while interfering with postsynaptic histamine, alpha-adrenergic, and muscarinic acetylcholine receptors [109]. Lastly, MAOIs inhibit monoamine oxidase enzyme, which metabolizes neurotransmitters such as serotonin, norepinephrine, and dopamine, thereby increasing their concentration in the brain [110].

Despite the proven efficacy of traditional antidepressants such as SSRIs, SNRIs, TCAs, and MAOIs, there are significant limitations associated with their use. One major issue is the delayed onset of therapeutic effects, as these medications typically require four to six weeks of continuous use before alleviating depressive symptoms [111]. This delay has been primarily attributed to the need for profound neuronal changes, such as synaptic plasticity and neurogenesis, fundamental processes for improving brain function that require time to manifest. In this regard, increased expression and signaling of BDNF, along with other factors like NGF and tyrosine kinase B (TrkB) receptors, have been directly linked to improved therapeutic responses to antidepressants. BDNF plays a crucial role in synaptic plasticity and response to both conventional and fast-acting antidepressants [112]. This delay in therapeutic response can prolong patient suffering and increase the risk of suicide, particularly in those with severe symptoms.

Another significant challenge in treating depression is adherence to therapy. Approximately 30% of patients discontinue treatment within the first month, and up to 60% do it within the first three months, often leading to worse long-term outcomes, such as increased relapse risk and reduced quality of life [113]. While various sociocultural factors, such as religious beliefs and stigma, as well as patient comorbidities, can influence non-adherence, the primary reasons for treatment discontinuation are related to the side effects of antidepressant medications [114].

Antidepressant side effects can be severe enough to induce treatment discontinuation. Common side effects include headaches, constipation, and weight gain, which affect a high percentage of patients. In some cases, treatment tolerability becomes a determining factor for discontinuation, as it is estimated that approximately 43% of patients with major depressive disorder stop taking their medications due to side effects [115]. Moreover, serious side effects, such as gastrointestinal bleeding, cardiovascular dysfunction, and liver function alterations, further limit the use of these medications, particularly older ones like TCAs and MAOIs. Studies have also highlighted an increased risk of gastrointestinal bleeding and cardiac rhythm disturbances with SSRIs, while SNRIs like venlafaxine are associated with a significantly higher risk of hypertension. Sexual dysfunction, affecting up to 80% of patients in clinical trials, is another major concern, as it interferes with quality of life and contributes to treatment dropout [116]. This challenge has driven research, primarily at the preclinical level, to develop new therapeutic alternatives that maintain the efficacy of current treatments while offering faster onset of action and fewer side effects, aiming to improve tolerability and adherence.

Currently, there has been growing interest in emerging therapies that offer faster and more effective options for treating depression, particularly in patients resistant to conventional treatments. Among these, ketamine—a dissociative anesthetic—has demonstrated rapid antidepressant effects, even in patients who do not respond to other treatments [108]. This drug, an ionotropic antagonist of the *N*-Methyl-d-Aspartate (NMDA) glutamate receptor, produces therapeutic effects within two hours following a single low-dose intravenous infusion, with effects lasting up to two weeks. By blocking NMDA receptors, ketamine reduces spontaneous neurotransmission, causing a temporary increase in BDNF levels. Although this BDNF increase returns to baseline within 24 h, the antidepressant effects persist for several days or even weeks [117]. However, its prolonged use presents limitations. While its safety profile is adequate at single anesthetic doses, repeated use at subanesthetic doses can cause side effects such as perceptual disturbances, euphoria, dizziness, and increased blood pressure, which generally subside within 80 min. Additionally, its potential for abuse, due to recreational use, complicates its adoption as a standard treatment [118]. These limitations underscore the need for continued research into safer alternatives while maintaining the rapid efficacy of ketamine.

In addition to ketamine, next-generation antidepressants, such as vilazodone and vortioxetine, are gaining relevance due to their ability to act on multiple serotonergic receptors. Vilazodone, a partial agonist of the 5-HT_1A_ receptor and an SSRI, has demonstrated a faster onset of therapeutic effects and fewer sexual side effects compared to other SSRIs. It has also proven beneficial for patients with major depression and high levels of anxiety [119]. Vortioxetine, on the other hand, has agonist properties at 5-HT_1A_ and 5-HT_1B_ receptors and antagonist effects at 5-HT_3_ and 5-HT_7_ receptors, contributing to faster action and improved side effects typical of SSRIs, such as nausea [116]. These characteristics make vilazodone and vortioxetine attractive therapeutic options, particularly for patients seeking treatments with fewer side effects and faster therapeutic onset.

Another emerging approach in the treatment of depression involves adjunct therapies, such as the combination of bupropion with conventional antidepressants. Bupropion, a drug belonging to the aminoketone class, inhibits the reuptake of NE and DA without affecting 5-HT, histamine, acetylcholine, or adrenaline receptors. This mechanism avoids significant sedation, cognitive disturbances, and anticholinergic effects [120]. While not a first-line option, its pharmacodynamics suggest it can enhance the effects of other treatments. Bupropion is generally well-tolerated, with side effects comparable to SSRIs, but it offers advantages such as not inducing weight gain—potentially leading to weight loss—and improving sexual function. These benefits are particularly relevant given that weight gain and sexual dysfunction are common adverse effects of both depression and other antidepressants [121]. While these options address specific aspects of depression, they are often limited by side effects and delayed onset. These challenges underscore the need for a more nuanced approach to treatment, particularly for patients who do not respond adequately to existing therapies. As research continues to advance, attention has increasingly turned toward emerging therapies that aim to bridge the gaps left by traditional treatments.

Given the need to improve existing pharmacological therapies for depression and anxiety disorders, researchers have begun to explore new alternatives that could be equally effective but without the short- and long-term side effects of traditional pharmacological therapies. In this context, probiotics have gained increasing attention due to their antidepressant and anxiolytic properties, observed in both clinical and preclinical studies. Emerging evidence suggests that probiotics may influence the MGB axis, modulating key neurotransmitters essential for mood regulation as well as cognitive and memory functions [4,5]. Several studies have documented the anxiolytic and antidepressant effects of probiotics in both animal models and humans [11,12]. While the exact mechanisms of their action are not yet fully understood, it has been proposed that probiotics may exert their effects through the modulation of inflammation and oxidative stress—processes closely linked to the etiology of mental disorders and mood disturbances [2,3,13]. However, further research is needed to elucidate the therapeutic effects of probiotics, particularly regarding their efficacy, tolerability, and long-term safety.

## 6. Experimental Evidence of Anxiolytic and Antidepressant Effects of Probiotics

The use of probiotics has recently increased, and pathological diseases have been alleviated by regulating the probiotic-mediated microbiota towards the MGB axis. Although research is still in its early stages, clinical and preclinical studies have demonstrated the potential of probiotics in the context of mental health by creating the foundation for applying preclinical models to people. Figure 2 describes the current mechanisms in the effect of probiotics on anxiety and depression considering the findings of preclinical and clinical studies.

Probiotic intervention can help treat possible etiopathogenic causes of depression and anxiety that also involve inflammation, neurotransmitters, the HPA axis, and epigenetic mechanisms because probiotics produce anti-inflammatory properties and help restore the levels of neurotransmitters involved in the onset of depression such as 5-HT, DA, NE, and GABA [122]. Since animal models are primarily employed to assess the safety of probiotic bacteria and potential mechanisms of action, the initial step in understanding the physiological action of probiotic strains is key to developing in vivo research to demonstrate their health benefits [123].

Studies in C57BL/6J male mice have shown that probiotics can reduce anxiety- and depression-like behaviors in standardized and validated models. In this regard, *Bifidobacterium* breve CCFM1025 showed a reduction in depression and anxiety behaviors in male mice during a 5-week treatment, regulating the HPA axis and increasing BDNF expression as well as 5-HT levels [124]. On the other hand, *Lactobacillus plantarum* ATCC 8014 in a study with male rats for 8 weeks showed a significant increase in antioxidant markers in serum and the amygdala as well as an increase in BDNF and 5-HT levels in the amygdala. This was associated with behavioral effects in the elevated plus maze and the forced swim tests [125]. *Lactobacillus plantarum* 90sk and *Bifidobacterium adolescentis* 150 in a study with male mice treated for 2 consecutive weeks showed a reduction in depression-like behavior in the forced swim test, with comparable effects to those produced by the antidepressant fluoxetine; these effects of probiotics appear to be associated with the ability of bacteria to synthesize GABA [126].

A preliminary study by Messaoudi et al. (2011) [127] examined the effects of a probiotic preparation containing *L. helveticus* R0052 and *B. longum* R0175 in rats dissolved in a 0.9% NaCl solution. On day 14 of treatment, a probe was placed inside the cages and, when the mice touched it, it gave them a mild electric shock. To support the anxiolytic properties of the compound, the results of the global stress/anxiety score were determined. The research team used the same formulation in a 30-day, randomized, double-blind, controlled study that included men and women participants. Participants had blood drawn as part of the pre-test medical checkup to ensure their biosafety parameters were within normal ranges. Because patients in the probiotic formulation group had reduced levels of somatization, sadness, and anger/hostility compared to patients in the control group over time, the overall severity index of the Hopkins Symptom Checklist (HSCL-90) was lower in patients in the probiotic formulation group. In addition, patients treated with probiotics had a lower self-blame score. They were reported to be more focused on problem solving than controls, indicating a difference between the two groups in emotional reactivity. These investigations showed that consumption of *B. longum* R0175 and *L. helveticus* R0052 reduced symptoms of anxiety and depression without causing side effects [127].

Similarly, research conducted by Ramalho et al. (2019) [128] provides evidence for the anxiolytic and antidepressant properties of probiotics. This was achieved through a series of tests to evaluate the potential effects of *L. lactis* subspecies cremoris LL95 in female C57BL/6 mice. To simulate the vulnerability of females to the onset of mood disorders, the study was conducted in vivo on 90-day-old female mice. The findings revealed that treatment with *L. lactis* LL95 had no adverse effects on any of the mice’s locomotor activities during the study or open-field trial. *L. lactis* LL95 has also demonstrated a free radical scavenging effect, suggesting that it may be a potential antioxidant therapy. Treatment with *L. lactis* LL95 also decreased depressive behavior in mice. Observations from this mouse model also indicate that oral supplementation with *L. lactis* LL95 can potentially reduce symptoms of anxiety and depression [128]. Other research shows that certain probiotics, specifically from the genera *Lactobacillus* and *Bifidobacterium*, are effective in reducing anxiety behaviors in various standardized tests such as the elevated plus maze test, open field test, light-dark box, and conditioned defensive burying test. It is suggested that probiotics may affect activity in areas of the brain that regulate the stress and anxiety response, such as the caudal solitary nucleus, periaqueductal gray matter, and central amygdala [129]. This implies that probiotics not only act at the gut level but also have an impact on neurochemistry and brain function related to anxiety.

In another study, the anxiolytic and antidepressant properties of the probiotic *Faecal-ibacterium prausnitzii* were investigated to measure how much the negative effect generated by chronic unpredictable mild stress (CUMS) was reduced in rats. The research was divided into two phases: the CUMS procedure period, which lasted from 3 to 6 weeks, and the recovery period, which lasted from 8 to 11 weeks. Several low-intensity stressors were applied to the rats. Throughout the 4-week trial, these stressors were repeated for 2 h each day for 1 week on a randomly assigned schedule. According to the findings, adding *F. prausnitzii* to a diet had both preventive and therapeutic effects on depression and anxiety brought on by CUMS. It prevented the effects of CUMS on the release of corticosterone, C-reactive protein, and IL-6 by increasing levels of SCFA and cytokines such as IL-10 in plasma. According to the results of this research, *F. prausnitzii* shows considerable potential as a probiotic agent [130].

On the other hand, meta-analyses have revealed that in preclinical studies, probiotics can reduce anxiety-like behavior in experimental subjects. An analysis of 22 preclinical studies showed that probiotics significantly reduced anxiety stage (Hedges’ *g* = −0.47), noting that the effect is more noticeable in animals that have developed the disorder. In this sense, the species *Lactobacillus rhamnosus* was identified as one of the strains with the greatest potential anxiolytic effect. The analysis of 14 clinical studies involving 1527 individuals found that only 3 studies reported that probiotics significantly reduced anxiety symptoms. Overall, the meta-analysis showed no significant anxiolytic effects in humans (Hedges’ *g* = −0.12), suggesting that probiotics do not have a clear therapeutic impact on anxiety in non-clinical populations [131]. It is suggested that probiotics may influence the immune system, which is a key component of the MGB axis. Preclinical studies have found that the anxiolytic effects of probiotics are accompanied by beneficial alterations in immune function. The microbiota stimulates a complex immune response and interacts with the intestinal barrier, and microbiota-free mice have been observed to have a compromised immune response. Therefore, probiotics could improve neuropsychiatric diseases by restoring microbiota-mediated immune activation to an adaptive level. In addition, it has been proposed that the observed anxiolytic effects could be due to the modulation of inflammation and the regulation of the HPA axis, although more research is needed to confirm these mechanisms [131,132].

Clinical studies on combination treatment with probiotics have demonstrated the effect of combining probiotic strains including *B. coagulans* Unique IS2, *L. rhamnosus* UBLR58, *B. lactis* UBLa70, *L. plantarum* UBLP40, *B. breve* UBBr01, and *B. infantis* UBBI01 for 28 days in young people subjected to stress with improvements in performance on stress tests and a significant reduction in serum cortisol levels [133]. On the other hand, multi-species mixing with *B. breve* CCFM1025, *B. longum* CCFM687, and *Pediococcus acidilactici* CCFM6432 for 4 weeks reported a significant improvement in depressive symptoms and gastrointestinal functions. Interestingly, these effects were not related to changes in the composition of the gut microbiota, suggesting a different mode of action [134].

The PROVIT study, a randomized controlled trial conducted in Austria, found that consuming a multi-strain probiotic for 4 weeks increased the relative abundance of *Ruminococcus gauvreauii* and *Coprococcus*. These strains are recognized for their ability to produce butyrate, an SCFA that possess immunoregulatory and anti-inflammatory properties. Butyrate plays a crucial role in maintaining the integrity of the intestinal epithelial barrier and preventing inflammation in the brain [135]. In addition, an eight-strain probiotic formulation containing bacteria such as *Bifidobacterium*, *Lactobacillus*, and *Lactococcus* has been shown to reduce symptoms of depression, anxiety, and stress in humans [136,137]. Table 1 includes preclinical studies describing the mechanism by which probiotics could act as anxiolytic or antidepressant agents.

It is crucial to recognize that changes in the composition of the gut microbiota induced by probiotics may be transient or even negligible in certain individuals, especially during short periods of intervention. There is considerable temporal variation for most major gut genera, with daily absolute fluctuations in abundance within individuals being much greater than between individuals, with up to 100-fold shifts over study periods [147]. This inherent variability underscores the complexity of achieving and recognizing meaningful microbial changes through probiotic interventions.

The systematic review and reanalysis by Ng et al. (2023) [147] in patients with major depressive disorder highlighted that no significant changes in gut microbiome profiles were observed after probiotic treatment in several studies. This observation raises important questions about the mechanisms of action of probiotics and suggests that therapeutic effects may not depend exclusively on detectable changes in microbial composition.

The interplay between resident and transient members of the microbial community is poorly defined, and it remains unclear to what extent a host’s autochthonous gut microbiota influences niche permeability to transient bacteria [148]. This complexity may explain why some people show resistance to probiotic-induced microbial changes.

The therapeutic effects may require a longer duration of treatment or specific strategies for colonization with strains. Probiotics have been used for decades to treat a variety of diseases; however, the rationale for their use continues to evolve [149]. The lack of observed changes in the microbiota challenges conventional assumptions about the mechanism of probiotic efficacy and suggests that beneficial effects may occur through several alternative pathways, such as colonization-independent mechanisms: probiotics may exert therapeutic effects without establishing permanent colonization, possibly through metabolites produced during intestinal transit or through direct host–microbe interactions. Functional modulation without taxonomic changes: probiotics can influence microbial function (gene expression, metabolite production) without significantly altering detectable taxonomic composition. Direct host effects: direct interactions with intestinal epithelial cells, immune cells, or the enteric nervous system, independent of microbial colonization. Subtle or localized microbial changes: changes in specific niches or minority populations that may not be detectable by conventional sequencing techniques [149].

The microbial communities are in symbiosis with the host, contribute to homeostasis, and regulate immune function, but dysbiosis of microbiota can lead to dysregulation of body functions [150]. On the other hand, the safety and efficacy of probiotic supplementation in oncology patients is a complex issue that requires careful consideration. Although probiotics are generally considered beneficial and have demonstrated efficacy in acute infectious diarrhea [151,152], the current literature shows significant limitations in the systematic reporting of adverse events. Probiotics can be a rare cause of sepsis [153], and cases have been reported where Lactobacillus and Bacillus caused infections leading to sepsis in patients, which is of particular concern due to immunosuppression in cancer patients undergoing chemotherapy [154]. However, due to different reporting methods, it was unclear whether adverse events were associated with probiotic infections, and no probiotic-related deaths were reported [155]. This lack of systematic adverse event reporting, as noted by the reviewer, limits the current literature’s ability to confidently address questions regarding the safety of probiotic interventions, especially in vulnerable populations such as cancer patients receiving radiotherapy, where some anecdotal reports suggest possible complications that warrant further investigation and more rigorous safety-monitoring protocols. This complexity suggests that the effects of probiotics may be more complex than previously thought. These limitations underscore the need to develop more sensitive and specific biomarkers for evaluating the efficacy of probiotics as well as personalized strategies based on the individual characteristics of the baseline microbiome. Future studies should consider longer intervention periods, functional analyses in addition to taxonomic profiling, and assessment of multiple signaling pathways within the microbiota–gut–brain axis to fully understand the mechanisms of probiotic action in mood disorders.

Considering these findings, it is suggested that probiotics can be used as adjuvant therapy for emotional or mood disorders, relieving depressive symptoms in participants with and without a clinical diagnosis. In this regard, possible mechanisms involved in the anxiolytic and antidepressant effects of probiotics include neurotransmitter modulation, as probiotics can influence the production and regulation of neurotransmitters such as 5-HT, GABA, NE, and DA, which are crucial for mood and emotional regulation. Reducing inflammation, probiotics have anti-inflammatory properties that can help mitigate systemic inflammation, a factor that has been linked to depression and anxiety. Another mechanism involved is interaction with the HPA axis. Several studies have reported that probiotics can affect HPA, which plays an important role in stress response and emotional regulation. Finally, maintaining the microbial diversity generated by probiotic treatment in the long term may be beneficial for the maintaining of mental health [122].

## 7. Conclusions

Probiotics are a promising intervention for anxiety disorders and depression. By modulating the MGB axis, these beneficial bacteria have demonstrated the ability to influence the regulation of essential mood-forming neurotransmitters such as 5-HT and GABA as well as reduce inflammation and oxidative stress, which are key factors in the etiology of mood disorders, with certain strains such as Lactobacillus and Bifidobacterium showing consistency between preclinical and clinical studies through the modulation of GABA and serotonin. However, research is still in its beginning, and there are important limitations that need to be addressed. Future research should focus on (1) methodological standardization with consensus protocols for dose (≥10^9^ CFU/day) and duration (at least 8 weeks); (2) development of microbiome biomarkers for therapeutic personalization; (3) combination studies with antidepressants to evaluate specific synergies; (4) clarification of pathways through targeted mechanistic studies; and (5) if probiotics exert strain-specific, dose-specific effects or vary by condition. For immediate clinical implementation, probiotics should be considered as adjunctive therapy in patients with gastrointestinal comorbidities, clinical practice guidelines for strain selection should be developed, and long-term follow-up protocols should be established. Successful integration into psychiatry requires interdisciplinary collaboration and targeted research addressing these specific elements to maximize the therapeutic potential of these microbiome-targeted interventions.

## 8. Future Directions

Future research should focus on developing large-scale controlled clinical trials that will allow for the establishment of standardized therapeutic protocols for the use of probiotics in anxiety and depression disorders. It is vital to determine the optimal doses, treatment duration, and the specific selection of the most effective probiotic strains for each clinical condition. Furthermore, a greater understanding of the underlying neurobiological mechanisms is needed, particularly in the bidirectional communication of the MGB axis and its influence on HPA axis regulation. Personalized therapy studies that consider the individual variability of the gut microbiome and its response to different probiotic interventions should be included. Likewise, it is necessary to explore combination therapies that integrate probiotics with other conventional pharmacological treatments and evaluate their long-term effects. Finally, further research is needed to develop specific biomarkers that allow for monitoring therapeutic efficacy and predicting individual response to probiotic treatments in the context of mental health.

## Figures and Tables

**Figure 1 biomedicines-13-01831-f001:**
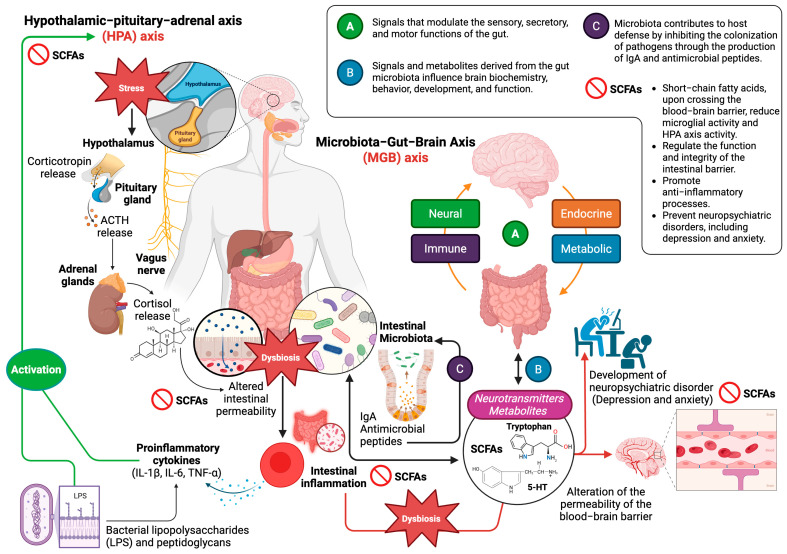
Communication axes and functions of the MGB axis. Schematic representation of the bidirectional relationship between the gut microbiota and the brain through the MGB axis. The main pathways involved are illustrated: vagus nerve; immune system; HPA axis; and microbial metabolites such as SCFAs, neurotransmitters, and gut hormones. The microbiota regulates functions such as gut and BBB integrity, immune system modulation, neuroinflammation, and neuromodulator synthesis, influencing individual physiology and behavior. This figure illustrates the role of dysbiosis and gut permeability in the development of neuropsychiatric disorders. Figure created with BioRender (https://www.biorender.com, accessed on 5 June 2025).

**Figure 2 biomedicines-13-01831-f002:**
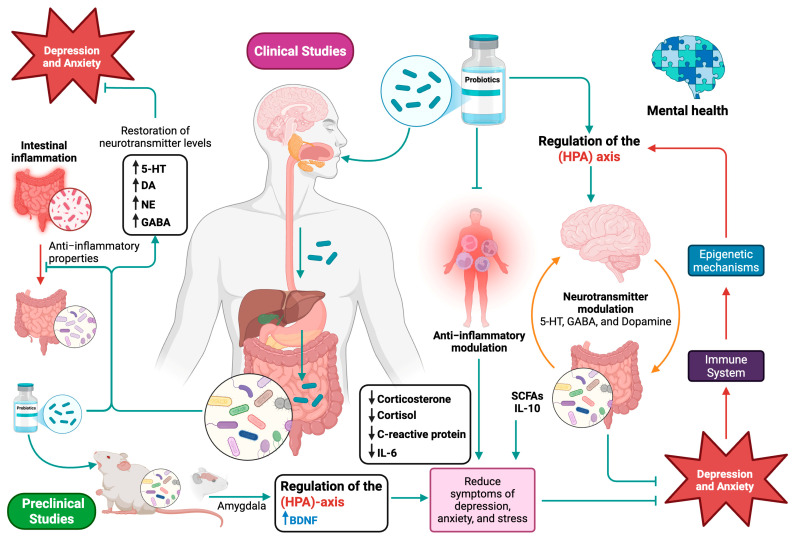
Mechanisms involved in the effect of probiotics on anxiety and depression. Schematic representation of the molecular and physiological mechanisms by which probiotics influence mental health through the MGB axis. Pathways such as neurotransmitter modulation (5-HT, GABA, DA, NE); HPA axis regulation; and reduction of systemic inflammation and metabolite production, such as SCFAs, are visualized. Key findings from animal model studies and human clinical trials in which probiotics demonstrate anxiolytic and antidepressant effects are summarized. Behavioral improvements, reduction of inflammatory markers, and decreased cortisol are indicated, with therapeutic implications for emotional disorders. Figure created with BioRender (https://www.biorender.com).

**Table 1 biomedicines-13-01831-t001:** Effects of probiotics on animal models of anxiety and depression.

Experimental Subject	Treatment and Dosage	Time of Treatment and Route of Administration	Animal Model	Effects of Treatment	Mechanism Involved	References
**Male Wistar rats**	*Lactobacillus helveticus* R0052 y *Bifidobacterium longum* R0175 (1 × 10^9^ CFU). Dosage: 250 mg/rat per day.	14 days. Orally gavaged.	Defensive flight test under stress conditions. Conditioned defensive burying.	Reduction in stress and anxiety scores in the conditioned defensive burying test, presenting a lower stress response compared to rats that received vehicle (placebo).	Modulation of the HPA axis, which regulates the stress response. Changes in the gut microbiota influence the release of neuroendocrine mediators, which could reduce the exaggerated response of the HPA axis to stressful stimuli.	[127]
**Pregnant Sprague Dawley**	*Bifidobacterium infantis* 35624 (1 × 10^10^ CFU per capsule). Citalopram 30 mg/kg.	40 days. Orally gavaged.	Maternal separation test. Forced swim test (FST). LPS.	*Bifidobacterium infantis* reduced IL-6 levels, immobility in FST, restored NE levels in the brainstem, decreased CRF expression in the amygdala, and had a modulatory effect on HPA. Citalopram restored 5-HT levels, reduced immobility in FST, increased NE levels in locus coeruleus, and reduced CRF levels in amygdala.	Modulation of the immune response by reducing IL-6 levels, restores HPA axis function by reducing CRF levels in the amygdala and normalizing NE levels in the brainstem.	[138]
**Adult male BALB/c mice**	*Lactobacillus rhamnosus* JB-1 (1 × 10^9^ CFU per capsule).	28 days. Orally gavaged.	Hypothermia-induced stress. Elevated plus maze (EPM). FST. Tail suspension test (TST). Fear conditioning test.	Reduction of depressive behaviors in FST alone. Decreased stress response. Improvement in behaviors suggestive of anxiety. Increased fear memory. Neurochemical alterations in GABA receptors.	Regulation of the GABAergic System and of the HPA axis. Production or regulation of neuroactive molecules such as GABA or modulation of other neurotransmitters and neuropeptides that act at the peripheral level and impact neuroendocrine and neurochemical pathways. This modulation is done, in part, through the activation of the vagus nerve, which transmits signals from the gut to the brain.	[139]
**Pregnant female Sprague-Dawley rats**	*Bacillus coagulans* Unique IS-2 (2 × 10^9^ CFU per capsule).	6 weeks. Orally gavaged.	Sucrose preference test (SPT). FST. EPM. Open field test (OFT).	*Bacillus coagulans* Unique IS-2 reduces behaviors associated with anxiety and depression. Normalizes 5-HT and BDNF levels. Reduces CRP, TNF-α, and IL-1β levels. Increases levels of SCFAs such as acetate, propionate, and butyrate.	The production of SCFAs modulates mitochondrial function, reduces inflammation, and strengthens the integrity of the intestinal barrier, preventing systemic inflammation that affects the brain. It also restores the balance of the intestinal microbiota, increasing beneficial bacteria such as *Firmicutes* and decreasing *Proteobacteria*, which favors the regulation of 5-HT and BDNF.	[140]
**Male BALB/cOlaHsd (BALB/c) mice**	*Bifidobacterium longum* 1714 and *Bifidobacterium breve* 1205 (1 × 10^9^ CFU/mL). Escitalopram 20 mg/kg.	6 weeks. Orally gavaged.	Hypothermia-induced stress. Marble burying. EPM. OFT. TST. FST.	*Bifidobacterium breve* 1205 increased time in open arm, decreased weight gain. Reduced stress-associated defecation and decreased marble burying time. *Bifidobacterium longum* 1714 had no effect on EPM, but it did reduce hypothermia and immobility time in FST and TST. Escitalopram reduced spleen weight, corticosterone levels, immobility in TST, and stress-induced hypothermia.	Modulation of inflammation, modulation of the serotonergic system, impact on neuroendocrine and inflammatory signaling.	[141]
**ICR mice**	Fluoxetine 10 mg/kg. *Lactobacillus* (1 × 10^4^ CFU/mL daily). *Lactobacillus* (1 × 10^8^ CFU/mL daily). *Bifidobacterium* (1 × 10^4^ CFU/mL daily). *Bifidobacterium* (1 × 10^8^ CFU/mL daily). *Lactobacillus* and *Bifidobacterium* (1 × 10^4^ CFU/mL daily). *Lactobacillus* and *Bifidobacterium* (1 × 10^8^ CFU/mL daily).	28 days. Orally gavaged.	CUMS. SPT. OFT. FST. TST. Hot plate test.	High doses of probiotics, as well as fluoxetine, restored sucrose preference. High doses of probiotics and fluoxetine decreased FST and TST immobility time. High doses of probiotics increased activity in central areas and the number of cross-sections in OFT. High doses of probiotics also helped reduce pain sensitivity.	Modulation of MGB, increased levels of 5-HT, GABA, and BDNF. Inhibition of the kynurenine pathway and increased expression of tryptophan hydroxylase 1 (TPH1). Additionally, they increased the activity of GABA receptors in brain regions related to mood and anxiety regulation. They reduced oxidative stress and systemic inflammation, helping to maintain the integrity of the intestinal barrier and prevent inflammatory processes associated with depression.	[142]
**Adult female C57BL/6J mice**	Fluoxetine 20 mg/kg. *Lactobacillus plantarum* CR12 and *Lactobacillus plantarum* ST-1 (1 × 10^9^ CFU). Dosage: 200 µL.	14 days. Orally gavaged.	SPT. FST. TST. OFT. Morris water maze.	*L. plantarum* CR12 and fluoxetine reduced depressive and anxious behavior, increased spatial memory, and reduced obsessive-ritual behaviors. Increased production of SCFAs, particularly butyrate, which are involved in regulating brain function and inflammation.	Modulation of MGB, production of S butyrate that act as mediators that cross the intestinal barrier and modulate neuroinflammation, reducing proinflammatory activation of microglia in the hippocampus. Improved intestinal barrier integrity through the restoration of proteins such as ZO-1, which reduces intestinal permeability and, consequently, the entry of proinflammatory molecules. L. plantarum CR12 promotes neuroprotection by reducing neuroinflammation.	[143]
**Fischer male rats** **Long Evans female rats**	Probiotics mixture (*Lactobacillus helveticus* LA 102, *Bifidobacterium longum* LA 101, *Lactococcus lactis* LA 103 and *Streptococcus thermophilus* LA 104 (1 × 10^9^ CFU). Dosage: 0.5 mL.	Fischer rats: 5 weeks. Long Evans rats: 9 weeks. Orally gavaged.	Maternal deprivation in Long Evans rats. Novel object test. Light-dark box test. EPM. OFT. FST.	Reduction of anxiety- and depression-related behaviors. Modification of the gut microbiota. Modification of the expression of inflammatory markers and binding proteins in the gut.	Modulation of the gut microbiota, which influences gut barrier function and reduces inflammation. Changes in the expression of epithelial junction proteins and inflammatory markers. Modification of the production and regulation of metabolites such as 21-deoxycortisol, which modulates brain monoamine levels.	[144]
**Male ICR mice**	Fluoxetine 10 mg/kg. Probiotic mixture (*Lactobacillus plantarum* LP3, *L. rhamnosus* LR5, *Bifidobacterium lactis* BL3, *B. breve* BR3, and *Pediococcus pentosaceus* PP1) 2 × 10^8^ CFU. Dosage: 500 µL.	8 weeks. Orally gavaged.	FST. TST. OFT. EPM.	Reduction of immobility in FST and TST. Increased locomotor activity without significant differences. Reduction in corticosterone levels. Restoration of the intestinal microbiota.	Modulation of the composition of the intestinal microbiota, which reduces inflammation and strengthens the intestinal barrier, preventing excessive activation of the immune system. This, in turn, helps regulate the MGB axis, decreasing the stress response and corticosterone levels, in addition to influencing the production of neurotransmitters related to mood.	[145]
**Swiss male mice**	*Lactobacillus plantarum* 286 *Lactobacillus plantarum* 81 (1 × 10^9^ CFU). Dosage: 0.1 mL.	30 days. Orally gavaged.	OFT. FST. Plus maze discriminative avoidance test (PM-DAT).	*Lactobacillus plantarum* 286 does not alter locomotor activity, memory, or learning in mice. It also showed anxiolytic and antidepressant effects.	Modulation of the MGB axis and GABAergic activity by the effects observed in EPM.	[146]

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
