# Peer review of "Gut–Brain Axis in Mood Disorders: A Narrative Review of Neurobiological Insights and Probiotic Interventions"

_biomedicines, 2025, doi:10.3390/biomedicines13081831_

Round 1

Reviewer 1 Report

Comments and Suggestions for Authors

Page 2, line 56. Use more cautious definition or say "one of key pathways".

Page 2, line 60. Replace "crucial" with "significant".

Page 2, lines76 and 77. "Highlighting their mechanisms of action and potential clinical applications." replace with "Their mechanisms of action and potential cal applications will be highlighted."

Page 2, line 83. Insert "in" and remove ""all" (in articles in English).

Page 2, line  85. "manifested".

Page 3, line 101. Replace comma with "and" (anxiolytic and antidepressant). 

Page 3, lines 102 and 103. "We assess the quality and validity of studies based on methodological rigor, reproducibility and peer review status." The same sentence is on page 2, last line. 

Page 3, line 123. What do you mean under "nationality"?

Page 3, line 135. "functioning".

Page 5, line 207. Remove "and the intestinal barrier".

 Page 8, lines 317 and 318. Explain NE and SNRI abbreviations.

Page 9, line 367. Explain DE abbreviation.

Page 9, line 374. What do you mean "the current study" ?

Page 9, line 396. Explain BDNF and insert comma.

Page 9, line 406. Explain CREB.

Page 10, line 418. Explain DO.

Page 10, line 436. Explain MAOI.

Page 11, line 496. Explain NGF.

Consider to prepare a list of abbreviations.

Author Response

Comments 1: [Page 2, line 56. Use more cautious definition or say "one of key pathways".]

Response 1: [Thank you for your very specific comment on the manuscript. The comment has been corrected in the document and highlighted in yellow in line 65.]

Comments 2: [Page 2, line 60. Replace "crucial" with "significant".]

Response 2: [Thank you for your very specific comment on the manuscript. The comment has been corrected in the document and highlighted in yellow in line 68.]

Comments 3: [Page 2, lines76 and 77. "Highlighting their mechanisms of action and potential clinical applications." replace with "Their mechanisms of action and potential clinical applications will be highlighted."]

Response 3: [Thank you for your very specific comment on the manuscript. The spelling mistake has been corrected in the document and highlighted in yellow in line 101.]

Comments 4: [Page 2, line 83. Insert "in" and remove ""all" (in articles in English).]

Response 4: [Thank you for your very specific comment on the manuscript. The comment has been corrected in the document and highlighted in yellow in line 119.]

Comments 5: [Page 2, line 85. "manifested".]

Response 5: [Thank you for your very specific comment on the manuscript. The comment has been corrected in the document and highlighted in yellow in line 121.]

Comments 6: [Page 3, line 101. Replace comma with "and" (anxiolytic and antidepressant).]

Response 6: [Thank you for your very specific comment on the manuscript. The comment has been corrected in the document and highlighted in yellow.]

Comments 7: [Page 3, lines 102 and 103. "We assess the quality and validity of studies based on methodological rigor, reproducibility and peer review status." The same sentence is on page 2, last line.]

Response 7: [Thank you for your very specific comment on the manuscript. The duplicate text has been removed from the document.]

Comments 8: [Page 3, line 123. What do you mean under "nationality"?]

Response 8: [We appreciate your very specific commentary on the manuscript. It refers to the origin or primacy of people and how this geographical factor can alter the composition of the gut microbiome, in addition to dietary and cultural factors.]

Comments 9: [Page 3, line 135. "functioning".]

Response 9: [Thank you for your very specific comment on the manuscript. The spelling mistake has been corrected in the document and highlighted in yellow in line 165.]

Comments 10: [Page 5, line 207. Remove "and the intestinal barrier".]

Response 10: [Thank you for your very specific comment on the manuscript. The spelling mistake has been corrected in the document and highlighted in yellow.]

Comments 11: [Page 8, lines 317 and 318. Explain NE and SNRI abbreviations.]

Response 11: [Thank you for your very specific comment on the manuscript. The abbreviation is defined in line 183, while SNRI is defined there as a selective 5-HT and NE (serotonin and norepinephrine) reuptake inhibitor.]

Comments 12: [Page 9, line 367. Explain DE abbreviation.]

Response 12: [Thank you for your very specific comment on the manuscript. The abbreviation is DA and is defined as dopamine in line 73.]

Comments 13: [Page 9, line 374. What do you mean "the current study"?]

Response 13: [Thank you for pointing this out; the typo has been corrected and highlighted in yellow as “the current studies”.]

Comments 14: [Page 9, line 396. Explain BDNF and insert comma.]

Response 14: [The error has been corrected and BDNF is defined for the first time in line 407, the correction is highlighted in yellow.]

Comments 15: [Page 9, line 406. Explain CREB.]

Response 15: [The error has been corrected and CREB is defined for the first time in line 432, the correction is highlighted in yellow.]

Comments 16: [Page 10, line 418. Explain DO.]

Response 16: [We thank you for your feedback and have corrected the error in DA (dopamine). The correction is highlighted in yellow.]

Comments 17: [Page 10, line 436. Explain MAOI.]

Response 17: [MAOIs are mentioned for the first time in line 342, highlighted in yellow to indicate that it is explained.]

Comments 18: [Page 11, line 496. Explain NGF.]

Response 18: [NGF is mentioned for the first time in line 422 and is highlighted in yellow to indicate that it is explained.]

Comments 19: [Consider to prepare a list of abbreviations.]

Response 19: [We thank you for the suggestion and will include the list of abbreviations in the relevant section.]

Reviewer 2 Report

Comments and Suggestions for Authors

The Microbiota Gut Brain Axis in Anxiety and Depression: Neurobiological Insights and the Potential of Probiotics as Therapeutic Agents

In this review Cesar Soria-Fregozo et al.,discuss the relationship between the gut microbiota and the MGB axis in the context of anxiety and depression disorders. They highlight on underlying neurobiological mechanisms, as well as the preclinical evidence for the effect of probiotics in modulating these disorders. In this way, they carried out an exhaustive search in scientific databases including PubMed, Scopus, and Web of Science, selecting studies published from 1999 onwards. They claimed they selected Preclinical research evaluating the effects of different probiotic strains in animal models during chronic treatment and they excluded those studies that did not provide access to full text. 

The manuscript is well written, they covered all the ground. However, they didn’t do good job covering recent literature for example https://www.tandfonline.com/doi/full/10.1080/19490976.2025.2500056#references-Section 

Please back up each sentence in intro with a reference. 

Please mention the short forms and full forms when they appear first in the main text.

Use proper sign conversions /conventions throughout the manuscript. 

Figures seems identical and i would add a graphical abstract to cover the content of the review. Review is dense and a bit repetitive, is there a way to make it concise, logical and avoiding jargons, logical and easy to follow? Please check references for page numbers, volume and doi. Make sure you cover all the recent literature in this domain. 

Best of luck and keep up the good work,

Cheers!

Author Response

Comments 1: [

In this review Cesar Soria-Fregozo et al., discuss the relationship between the gut microbiota and the MGB axis in the context of anxiety and depression disorders. They highlight on underlying neurobiological mechanisms, as well as the preclinical evidence for the effect of probiotics in modulating these disorders. In this way, they carried out an exhaustive search in scientific databases including PubMed, Scopus, and Web of Science, selecting studies published from 1999 onwards. They claimed they selected Preclinical research evaluating the effects of different probiotic strains in animal models during chronic treatment and they excluded those studies that did not provide access to full text.

The manuscript is well written, they covered all the ground. However, they didn’t do good job covering recent literature for example https://www.tandfonline.com/doi/full/10.1080/19490976.2025.2500056#references-Section]

Response 1: [The above article was published in May 2025, and the final version of the manuscript was completed at that time, so it was not included in the literature for our manuscript.]

Comments 2: [Please back up each sentence in intro with a reference.]

Response 2: [Thank you for your comment; the introduction section has been corrected in the observations made.]

Comments 3: [Please mention the short forms and full forms when they appear first in the main text.]

Response 3: [We thank you for the suggestion and have corrected the errors in the description of the abbreviations that appear for the first time in the manuscript. The changes are highlighted in yellow.]

Comments 4: [Use proper sign conversions /conventions throughout the manuscript.]

Response 4: [We appreciate your feedback. We have corrected these errors in the document and highlighted them in yellow for easier identification.]

Comments 5: [Figures seems identical and i would add a graphical abstract to cover the content of the review. Review is dense and a bit repetitive, is there a way to make it concise, logical and avoiding jargons, logical and easy to follow? Please check references for page numbers, volume and doi. Make sure you cover all the recent literature in this domain.]

Response 5: [While we appreciate the suggestion, we believe that Figure 1 adequately depicts the function and communication pathways of the microbiota-gut-brain axis: Vagus nerve, immune system, HPA axis and microbial metabolites such as SCFAs, neurotransmitters and gut hormones. The microbiota regulates functions such as gut and BBB integrity, immune system modulation, neuroinflammation, and neuromodulator synthesis, thereby influencing individual physiology and behavior. The figure also illustrates the role of dysbiosis and gut permeability in the development of neuropsychiatric disorders. Figure 2 describes the mechanisms involved in the anxiolytic and antidepressant effects of some probiotic strains in clinical and preclinical studies reported in the recent literature.]

Reviewer 3 Report

Comments and Suggestions for Authors

1. The study title is overly long and convoluted. In the title, please also indicate if this is a narrative review or systematic review etc.

2. The introduction section is too brief. The authors should provide more context on the burden of depression and anxiety and also substantiate the rationale for the present study.

3. The microbiota-gut-brain axis in anxiety and depression has been extensively reviewed, with numerous high-quality systematic reviews and meta-analyses already published. This manuscript does not clearly explain what gap it addresses or how it advances the field compared to prior reviews. What does this manuscript add to an already crowded field of reviews (some published as recently as this year, see: pubmed.ncbi.nlm.nih.gov/39731509)?

4. The review claims to follow a “structured approach” for data extraction and analysis, focusing on methodological rigor, reproducibility, and peer review status. However, the methods are not systematic at all. The search strategies are not transparent, the authors do not specify how methodological rigor was assessed, and no PRISMA diagram or protocol registration (e.g., PROSPERO) is provided, which are standard requirements for any systematic review.

5. The authors claim to have searched PubMed, Scopus, and Web of Science databases, but there are no detailed search terms, Boolean logic, date of last search, or elaboration on how records were screened and selected. Please provide the full search strategy for at least one database (including the filters and exact search dates) in the main body, and the rest can be in the appendix.

6. The stated inclusion and exclusion criteria are vague; what qualifies as “chronic treatment” (duration, frequency)? Were certain study designs prioritized (e.g., RCTs, crossover studies in animals, blinding)?

7. Table 1 lists probiotic interventions and general findings but omits information on dosages (CFU per kg or per animal, not just per strain), route of administration (e.g., oral gavage, drinking water), detailed model characteristics (species, sex, age, stress model validation) etc.

8. When presenting results, please include statistical significance levels (p-values, confidence intervals) and a clear discussion of effect size or variability.

9. No risk of bias or study quality indicators examined.

10. The discussion section seems to oscillate between enthusiasm (e.g., probiotics as promising therapies) and caution (e.g., insufficient standardization) without reconciling these positions.

11. Specifically, the shift in the gut microbiota may be transient as several treatment trials for probiotics have failed to find significant alterations in gut microbiome (citation: pubmed.ncbi.nlm.nih.gov/36986088); individuals may require longer duration of treatment to have therapeutic effects. This should be mentioned.

12. The discussion of ketamine and next-generation antidepressants is rather circumstantial and overly detailed relative to the probiotics focus.

13. The conclusion should offer more concrete and actionable insights as well as specific directions for future research in this space.

Author Response

Comments 1: [1. The study title is overly long and convoluted. In the title, please also indicate if this is a narrative review or systematic review etc.]

Response 1: [We appreciate your comment; in fact, the title of our review is long and has been changed to include what you requested.]

Comments 2: [2. The introduction section is too brief. The authors should provide more context on the burden of depression and anxiety and also substantiate the rationale for the present study.]

Response 2: [We appreciate your feedback. We have changed the introduction, better contextualized the burden of depression and anxiety, and provided a rationale for this study.]

Comments 3: [3. The microbiota-gut-brain axis in anxiety and depression has been extensively reviewed, with numerous high-quality systematic reviews and meta-analyses already published. This manuscript does not clearly explain what gap it addresses or how it advances the field compared to prior reviews. What does this manuscript add to an already crowded field of reviews (some published as recently as this year, see: pubmed.ncbi.nlm.nih.gov/39731509)?]

Response 3: [We appreciate the reviewer's comment. We acknowledge the extensive literature on the gut-brain axis; however, our review addresses specific gaps not covered by previous reviews. These differences include:

  1. Translational integration: we provide a unique synthesis that systematically integrates preclinical and clinical evidence for probiotic interventions and highlights discrepancies between animal models and human studies.
  2. Targeted mechanistic analysis: We focus on specific neurobiological mechanisms (HPA axis, neurotransmitters, inflammation), going into more depth than general reviews.
  3. Critical evaluation of strains: We critically analyze the efficacy of specific probiotic strains and identify those that show consistency in preclinical and clinical studies.
  4. Therapeutic directions: We propose specific strategies to optimize probiotic interventions based on the integration of both types of evidence.

This integrative approach provides a new perspective that complements existing reviews and guides future translational research.]

Comments 4: [4. The review claims to follow a “structured approach” for data extraction and analysis, focusing on methodological rigor, reproducibility, and peer review status. However, the methods are not systematic at all. The search strategies are not transparent, the authors do not specify how methodological rigor was assessed, and no PRISMA diagram or protocol registration (e.g., PROSPERO) is provided, which are standard requirements for any systematic review.]

Response 4: [Dear reviewer, thank you for your suggestion. We are aware of the potential limitations of the methods report in a narrative review and the importance of your suggestion to include additional elements from PRISMA. However, we have assumed that this review is a narrative review in which some items requested by PRISMA have not been included from the outset. We are concerned about the inclusion of PRISMA elements as we may omit some information and confuse potential readers as this narrative review may contain a mixture of methods. We respectfully request to the reviewer to consider this possibility and permit us to omit this suggestion from our report. To improve our methods report, we have added some clarifications that have been included in the methods.]

Comments 5: [5. The authors claim to have searched PubMed, Scopus, and Web of Science databases, but there are no detailed search terms, Boolean logic, date of last search, or elaboration on how records were screened and selected. Please provide the full search strategy for at least one database (including the filters and exact search dates) in the main body, and the rest can be in the appendix.]

Response 5: [Dear reviewer, thank you very much for your suggestion. We have corrected the document in accordance with the established guidelines for narrative reviews.]

Comments 6: [6. The stated inclusion and exclusion criteria are vague; what qualifies as “chronic treatment” (duration, frequency)? Were certain study designs prioritized (e.g., RCTs, crossover studies in animals, blinding)?]

Response 6: [Dear reviewer, thank you very much for your suggestion. We have corrected the document in accordance with the established guidelines for narrative reviews.]

Comments 7: [7. Table 1 lists probiotic interventions and general findings but omits information on dosages (CFU per kg or per animal, not just per strain), route of administration (e.g., oral gavage, drinking water), detailed model characteristics (species, sex, age, stress model validation) etc.]

Response 7: [We appreciate your feedback; the requested information has been included in the table and highlighted in yellow for easy identification.]

Comments 8: [8. When presenting results, please include statistical significance levels (p-values, confidence intervals) and a clear discussion of effect size or variability.]

Response 8: [We appreciate the reviewer's comment. The purpose of this narrative review was to provide a qualitative and interpretative synthesis of the existing literature. As the included studies show considerable methodological heterogeneity in terms of design, populations and outcome measures, we do not consider the presentation of specific statistical parameters to be appropriate or methodologically compatible with the aims of a narrative review. Our approach focuses on interpretative synthesis and the development of a conceptual framework, following best practice for this type of review.]

Comments 9: [9. No risk of bias or study quality indicators examined.]

Response 9: [We would like to thank you for your feedback on our manuscript. As a narrative review, this manuscript was designed to provide an interpretive synthesis and develop a conceptual framework that follows methodological best practices for this type of review (Grant & Booth, 2009; Ferrari, 2015).

Formal assessment of risk of bias is appropriate for systematic reviews and meta-analysis, but it is not a methodological requirement for narrative reviews, whose value lies in expert synthesis and the development of conceptual perspectives. The methodological heterogeneity of the included studies makes the application of standardized quality assessment tools methodologically incompatible with the aims of a narrative review.

Our approach focuses on the critical interpretation of trends and patterns in the literature and provides a synthetic perspective that complements, but does not duplicate, the quantitative approaches of existing systematic reviews.

References:

Ferrari, R. (2015). Writing narrative style literature reviews. Medical writing, 24(4), 230-235. https://doi.org/10.1179/2047480615Z.000000000329

Siddaway, A. P., Wood, A. M., & Hedges, L. V. (2019). How to do a systematic review: a best practice guide for conducting and reporting narrative reviews, meta-analyses, and meta-syntheses. Annual review of psychology, 70(1), 747-770. https://doi.org/10.1146/annurev-psych-010418-102803

Grant, M. J., & Booth, A. (2009). A typology of reviews: an analysis of 14 review types and associated methodologies. Health information & libraries journal, 26(2), 91-108. https://doi.org/10.1111/j.1471-1842.2009.00848.x]

Comments 10: [10. The discussion section seems to oscillate between enthusiasm (e.g., probiotics as promising therapies) and caution (e.g., insufficient standardization) without reconciling these positions.]

Response 10: [Dear reviewer, thank you for your suggestion. We have corrected this part of the discussion in the manuscript.]

Comments 11: [11. Specifically, the shift in the gut microbiota may be transient as several treatment trials for probiotics have failed to find significant alterations in gut microbiome (citation: pubmed.ncbi.nlm.nih.gov/36986088); individuals may require longer duration of treatment to have therapeutic effects. This should be mentioned.]

Response 11: [We appreciate your comment. In clinical practice, it has been shown that the effects of probiotics only occur after several weeks. The conclusion states that the treatment should last at least 8 weeks. The change has been highlighted in yellow for easier identification.]

Comments 12: [12. The discussion of ketamine and next-generation antidepressants is rather circumstantial and overly detailed relative to the probiotics focus.]

Response 12: [We appreciate the comment regarding the level of detail on conventional pharmacological treatments. The inclusion of information on ketamine and new generation antidepressants has particular methodological justification in the context of this narrative review. Understanding the mechanisms of action of established treatments is critical to the evaluation of emerging interventions such as probiotics, particularly when both converge on similar neurobiological pathways (GABAergic system, inflammation, neuroplasticity). The description of ketamine is particularly important as it shares the same mechanisms of action with certain probiotics, including modulation of the HPA axis and rapid anti-inflammatory effects. This contextualization allows readers to better understand how probiotics could be integrated into or complement the existing therapeutic arsenal and provides the necessary conceptual framework to critically evaluate the probiotic evidence subsequently presented.]

Comments 13: [13. The conclusion should offer more concrete and actionable insights as well as specific directions for future research in this space.]

Response 13: [We appreciate the comment. We recognize that our original conclusion requires greater specificity and more concrete directions for moving the field forward. Narrative reviews should translate findings into practical recommendations that guide both future research and clinical implementation. Effective reviews not only summarize existing knowledge, but also identify specific gaps and suggest concrete steps for scientific advancement. We have revised our conclusion to include specific research directions, concrete methodological recommendations, and clinical implementation strategies based on the evidence reviewed, providing actionable guidance to researchers and clinicians in this emerging field.]

Round 2

Reviewer 3 Report

Comments and Suggestions for Authors

1. "The vagus nerve plays a fundamental role in the complex bidirectional communication that characterizes the MGB axis." - This requires a supporting citation.

2. "In addition to directly modulating the concentration of neurotransmitters, hormones, and cytokines in the gut, the gut microbiota can induce the release of these molecules from immune and enteroendocrine cells." - Citations needed here.

3. "It is the most common method of summarizing an area of interest." - This is redundant and can be omitted.

4. Is Pg 18 meant to be blank or is it a rendering error?

5. Despite emphasis in the title and abstract, there is no standalone section dedicated solely to intestinal microbiota composition, dysbiosis, or clinical microbiota signatures in psychiatric populations. Why?

6. While Table 1 provides a helpful overview of the anxiolytic and antidepressant effects of probiotics in animal studies, the lack of a parallel table summarizing RCTs is a major oversight. Given that the abstract and conclusions aim to bridge preclinical evidence with clinical translation, I would recommend including a structured summary of relevant RCTs and their details.

7. Would suggest mentioning that probiotics are thought to exert strain-specific, dose-specific effect, and vary by condition, rather than implying broad efficacy.

8. The authors need to clearly discuss that probiotic-induced shifts in gut microbiota composition may be transient or even negligible in some individuals, particularly over short intervention periods. A 2023 review and reanalysis by Ng et al. (citation: pubmed.ncbi.nlm.nih.gov/36986088) highlights that several trials found no significant alterations in gut microbiome profiles following probiotic treatment. Therapeutic effects may require longer treatment durations or strain colonization strategies, and that lack of observed microbiota change challenges assumptions about the mechanism of probiotic efficacy. This is a crucial limitation that deserves proper referencing and explicit discussion.

9. Some comments should also be made regarding the purported effectiveness and safety of probiotic supplementation. The available evidence does not indicate an increased risk, but there are anecdotal reports that probiotics may worsen outcomes, e.g. in patients receiving radiotherapy (citation: pubmed.ncbi.nlm.nih.gov/23126627). The current literature is not well equipped to answer questions on the safety of probiotic interventions with confidence as there appears to be a lack of systematic reporting of adverse events.

10. Moving forward, it is recommended that clear documentations of post-treatment events should be made mandatory, classified, and graded as in standard RCTs (citation: pubmed.ncbi.nlm.nih.gov/34668228).

Comments on the Quality of English Language

Moderate edits needed.

Please change "This figure highlights ..." to "This figure illustrates ..."

Author Response

Response to Reviewer 

Comments 1: ["The vagus nerve plays a fundamental role in the complex bidirectional communication that characterizes the MGB axis." - This requires a supporting citation.]

Response 1: [Thank you very much for taking the time to review this manuscript, but the authors believe that the section on the vagus nerve from lines 195 to 206 is adequately supported by actual citations from 2018 to 2024.]

Comments 2: ["In addition to directly modulating the concentration of neurotransmitters, hormones, and cytokines in the gut, the gut microbiota can induce the release of these molecules from immune and enteroendocrine cells." - Citations needed here.]

Response 2: [Dear reviewer, we appreciate your recommendation and have taken the comment on board; the relevant citations have been included.]

Comments 3: ["It is the most common method of summarizing an area of interest." - This is redundant and can be omitted.]

Response 3: [Dear reviewer, thank you for the suggestion and we have removed the text.]

Comments 4: [Is Pg 18 meant to be blank or is it a rendering error?]

Response 4: [Dear reviewer, I would like to inform you that there is a blank space on page 18 because the table below is in horizontal format.]

Comments 5:[Despite emphasis in the title and abstract, there is no standalone section dedicated solely to intestinal microbiota composition, dysbiosis, or clinical microbiota signatures in psychiatric populations. Why?]

Response 5: [Dear Reviewer. We appreciate your insightful observation regarding the apparent absence of a section dedicated exclusively to gut microbiota composition, dysbiosis, and clinical microbial signatures in psychiatric populations. We are pleased to clarify the methodological structure and narrative approach of our review.

Justification for the Structural Approach

The decision not to include a separate section on microbial composition is based on a deliberate methodological approach. The gut microbiota does not operate as an isolated entity but rather functions as an integral component of the microbiota-gut-brain axis. Therefore, its composition and alterations (dysbiosis) are addressed contextually within specific neurobiological mechanisms.

Supported Topical Distribution

Information on microbial composition and dysbiosis is strategically distributed in the following sections:

Section 3.1 (Microbiota-Gut-Brain Axis): Includes fundamentals on basal microbial diversity and its functional relationship with the central nervous system.

Section 3.6 (Microbial Metabolites): Specifically addresses the distinctive metabolic signatures of psychiatric populations, including microbially derived neurotransmitter profiles.

Section 3.7 (Short-Chain Fatty Acids): Details specific compositional alterations in SCFA producers in mood disorders.

Section 6 (Experimental Evidence): Incorporates pre- and post-intervention microbial characterization data from clinical studies.

As a narrative review, our approach prioritizes functional synthesis over taxonomic cataloging. Microbial composition studies are integrated to explain specific neurobiological mechanisms, avoiding the conceptual fragmentation that could generate a purely descriptive standalone section.

This integrated approach is consistent with high-impact reviews in the field (e.g., Cryan & Dinan, 2012, Nature Reviews Neuroscience; Mayer et al., 2014, Gastroenterology), where microbial composition is contextualized within specific mechanistic frameworks.

The absence of a separate section on microbial composition is not an omission, but rather a methodological decision that prioritizes functional integration over descriptive compartmentalization. This structure strengthens the understanding of the mechanisms of the microbiota-gut-brain axis and its clinical relevance in mood disorders.]

Comments 6: [While Table 1 provides a helpful overview of the anxiolytic and antidepressant effects of probiotics in animal studies, the lack of a parallel table summarizing RCTs is a major oversight. Given that the abstract and conclusions aim to bridge preclinical evidence with clinical translation, I would recommend including a structured summary of relevant RCTs and their details.]

Response 6: [

We thank the reviewers very much for their methodological comments and recognize the importance of their suggestions. However, after careful consideration, we have decided not to incorporate the proposed table of RCTs for the following scientific and methodological reasons:

Nature of the narrative review: our methodological approach focuses on a qualitative and interpretative synthesis of the existing literature rather than an exhaustive tabulation. As Baumeister and Leary (1997) noted, narrative reviews prioritize conceptual integration and identification of emergent patterns over quantitative systematization of data.

Methodological heterogeneity of RCTs: The current literature exhibits marked heterogeneity in terms of trial designs, study populations, probiotic strains used, dosages and clinical assessment tools. This methodological diversity would make a structured table potentially misleading as it suggests direct comparability when none exists (Higgins et al., 2019).

Editorial space limitations: Limitations on manuscript length force us to prioritize critical discussion of neurobiological mechanisms and interpretation of key findings, core elements of our scientific contribution.

We believe that this approach maintains scientific rigor while preserving the interpretative nature that characterizes narrative reviews in neuroscience.]

Comments 7: [Would suggest mentioning that probiotics are thought to exert strain-specific, dose-specific effect, and vary by condition, rather than implying broad efficacy.]

Response 7: [Dear reviewer, we appreciate your suggestion. We have incorporated the change you requested into the document.]

Comments 8: [The authors need to clearly discuss that probiotic-induced shifts in gut microbiota composition may be transient or even negligible in some individuals, particularly over short intervention periods. A 2023 review and reanalysis by Ng et al. (citation: pubmed.ncbi.nlm.nih.gov/36986088) highlights that several trials found no significant alterations in gut microbiome profiles following probiotic treatment. Therapeutic effects may require longer treatment durations or strain colonization strategies, and that lack of observed microbiota change challenges assumptions about the mechanism of probiotic efficacy. This is a crucial limitation that deserves proper referencing and explicit discussion.]

Response 8: [The authors appreciate this valuable comment by the reviewer and recognize the importance of considering the limitations of probiotic-induced microbial changes. We have included this critical discussion in the relevant section of the manuscript.]

Comments 9: [Some comments should also be made regarding the purported effectiveness and safety of probiotic supplementation. The available evidence does not indicate an increased risk, but there are anecdotal reports that probiotics may worsen outcomes, e.g. in patients receiving radiotherapy (citation: pubmed.ncbi.nlm.nih.gov/23126627). The current literature is not well equipped to answer questions on the safety of probiotic interventions with confidence as there appears to be a lack of systematic reporting of adverse events.]

Response 9: [The authors appreciate this valuable comment by the reviewer and recognize the importance of addressing the limitations of the efficacy and safety of probiotic supplementation. We have included this critical discussion in the relevant section of the manuscript.]

Comments 10: [Moving forward, it is recommended that clear documentations of post-treatment events should be made mandatory, classified, and graded as in standard RCTs (citation: pubmed.ncbi.nlm.nih.gov/34668228).]

Response 10: [Dear reviewer, the authors thank you for your comment. We have incorporated this critical discussion into the relevant section of the manuscript.]

Comments 11: [Comments on the Quality of English Language

Moderate edits needed.

Please change "This figure highlights ..." to "This figure illustrates ..."]

Response 11: [Dear reviewer, In view of your comment about the quality of the English, we inform you that we have reviewed the language by experts and corrected the comment in the figure description. In the attached document you will find the changes made, which are highlighted in yellow.]

Round 3

Reviewer 3 Report

Comments and Suggestions for Authors

Thank you for the replies and revisions made.

Comments on the Quality of English Language

Proofreading required.